# Advances in the Development of Positron Emission Tomography Tracers for Improved Detection of Differentiated Thyroid Cancer

**DOI:** 10.3390/cancers16071401

**Published:** 2024-04-02

**Authors:** Hannelore Iris Coerts, Bart de Keizer, Frederik Anton Verburg

**Affiliations:** 1Erasmus Medical Center, Department of Radiology and Nuclear Medicine, 3015 GD Rotterdam, The Netherlands; h.coerts@erasmusmc.nl; 2Department of Nuclear Medicine and Radiology, University Medical Center Utrecht, 3584 CX Utrecht, The Netherlands; b.dekeizer@umcutrecht.nl

**Keywords:** positron emission tomography, thyroid cancer, PSMA, FAPI, RGD, TFB

## Abstract

**Simple Summary:**

This review discusses new radiotracers for diagnosing and treating thyroid cancer. It focuses on PSMA-based tracers, FAPI-based tracers, RGD-based tracers, and 18F-TFB. These tracers show promise in detecting thyroid cancer, especially in challenging cases. They offer better accuracy and help in choosing the right treatment. Further research is needed, especially on the promising 18F-TFB and FAP-targeting tracers, to improve thyroid cancer staging and treatment outcomes.

**Abstract:**

Thyroid cancer poses a significant challenge in clinical management, necessitating precise diagnostic tools and treatment strategies for optimal patient outcomes. This review explores the evolving field of radiotracers in the diagnosis and management of thyroid cancer, focusing on prostate-specific membrane antigen (PSMA)-based radiotracers, fibroblast activation protein inhibitor (FAPI)-based radiotracers, Arg-Gly-Asp (RGD)-based radiotracers, and ^18^F-tetrafluoroborate (^18^F-TFB). PSMA-based radiotracers, initially developed for prostate cancer imaging, have shown promise in detecting thyroid cancer lesions; however, their detection rate is lower than ^18^F-FDG PET/CT. FAPI-based radiotracers, targeting fibroblast activation protein highly expressed in tumors, offer potential in the detection of lymph nodes and radioiodine-resistant metastases. RGD-based radiotracers, binding to integrin αvβ3 found on tumor cells and angiogenic blood vessels, demonstrate diagnostic accuracy in detecting radioiodine-resistant thyroid cancer metastases. ^18^F-TFB emerges as a promising PET tracer for imaging of lymph node metastases and recurrent DTC, offering advantages over traditional methods. Overall, these radiotracers show promise in enhancing diagnostic accuracy, patient stratification, and treatment selection in differentiated thyroid cancer, warranting further research and clinical validation. Given the promising staging capabilities of 18F-TFB and the efficacy of FAP-targeting tracers in advanced, potentially dedifferentiated cases, continued investigation in these domains is justified.

## 1. Introduction

Thyroid cancer is a common type of cancer that accounts for 3% of all cancer cases worldwide. Proper diagnosis and staging are important for determining the best treatment options and improving patient outcomes. Traditional imaging techniques like ultrasound and computed tomography (CT) have been useful for evaluating thyroid nodules and assessing thyroid cancer. However, using PET tracers like ^18^F-FDG PET/CT can provide more accurate results in detecting and characterizing thyroid cancer lesions. According to the American Thyroid Association (ATA) guidelines, advanced imaging studies like CT and MRI with intravenous contrast should be used alongside ultrasound for preoperative assessment in patients with suspected advanced thyroid disease [1]. The European Association of Nuclear Medicine (EANM) recommends ^18^F-FDG PET/CT for differentiated thyroid cancer (DTC) patients with rising Tg levels to aid in detecting recurrent/metastatic disease and determining the tumor’s biological behavior [2].

Follow-up DxWBS in combination with Tg measurement is useful for evaluating therapy response and identifying patients with iodine-avid metastatic disease. Brain MRI is recommended in advanced DTC cases due to the potential presence of brain metastases, especially in the absence of neurological signs or symptoms, as ^18^F-FDG PET/CT is not reliable in the brain due to high glucose metabolism. The most commonly used PET tracers are ^18^F-FDG and iodine-124 (^124^I). Iodine-124 is a radionuclide that emits positrons, making it suitable for NIS imaging with PET. However, ^124^I has some limitations: (1) its 4.2-day half-life and low positron yield lead to high radiation exposures, making it less desirable for diagnostic imaging; (2) its high-energy single photon emissions (0.6–1.7 MeV) can compromise image quality; and (3) its high positron emission energy results in maximum tissue travel of >6 mm before annihilation, negatively impacting imaging resolution. In clinical settings, ^18^F-FDG PET/CT proves valuable as a prognostic factor for RAI treatment response in metastatic DTC patients, with high ^18^F-FDG uptake serving as an independent negative prognostic factor for overall survival.

In recent times, much effort has been spent on trying out numerous new tracers for imaging in various clinical stages of DTC in order to advance imaging sciences, especially where ultrasound and ^18^F-FDG PET/CT are not sufficient. This has resulted in numerous publications on various situations, without a good overview of the performance of different tracers in similar situations currently available in the literature. Hence, this review aims to comprehensively evaluate the literature on new tracers for different clinical indications in thyroid cancer, such as detecting lymph node metastases, TENIS (thyroglobulin elevation/negative iodine scintigraphy) syndrome, recurrent or persistent disease, and distant metastases in DTC.

## 2. Prostate-Specific Membrane Antigen-Based Radiotracers for Differentiated Thyroid Cancer

Prostate-specific membrane antigen (PSMA) is a protein that was first discovered to be overproduced in prostate cancer cells. Recently, the United States Food and Drug Administration approved ^68^Ga-PSMA-11 PET imaging to detect the potential spread or recurrence of prostate cancer in patients. ^68^Ga-PSMA PET/CT is now being used in clinical practice and trials for prostate cancer diagnosis. Although PSMA expression has been reported in the tumor-associated endothelium of various malignancies, including colon, breast, and adrenocortical cancers, its expression in thyroid tumors has not been systematically studied. Recently, several studies found unexpected PSMA radiotracer uptake by thyroid tumors, including radioiodine-refractory (RAI-R) DTC, which has led to an interest in studying ^68^Ga-PSMA PET/CT for thyroid cancer.

PSMA is a type 2 transmembrane protein, and it facilitates endothelial cell invasion through the extracellular matrix by interacting with the cytoskeleton via integrin signaling and actin-binding protein Filamin A. Therefore, it is overexpressed on the endothelial cells of the tumor neovasculature in several solid [3]. It has been found through functional studies that PSMA plays a role in tumor angiogenesis and is part of a self-regulating loop that involves β1-integrin and p21-activated kinase 1 (PAK1). This has led to the hypothesis that ^68^Ga-PSMA PET/CT could be useful in detecting thyroid cancer. ^68^Ga-PSMA PET/CT could aid in detecting a recurrence in RAI-R patients, identifying metastases in TENIS patients, and selecting patients who could benefit from PSMA-targeted radionuclide therapy (^177^Lu-PSMA-617), particularly those with metastatic dedifferentiated thyroid cancer.

Several preclinical studies have examined the expression of PSMA in thyroid tissue, various types of thyroid cancer, lymph node metastases, in vivo small animal models, its potential role as a biomarker for tumor aggressiveness, and its correlation with patients’ RAI positivity or negativity. Bychkov et al. (2017) conducted a study of 267 individuals and found that PSMA expression was commonly present in thyroid tumor microvessels, but absent in benign tissue [4]. PSMA expression varied significantly, ranging from 19% in benign tumors to over 50% in thyroid cancer, with a high degree of heterogeneity. PSMA expression was not found to be directly associated with endothelial cell proliferation, as confirmed by immunostaining with the endothelial marker CD105. Heitkotter and colleagues also reported similar outcomes, where overall PSMA expression was significantly higher in malignant tumors (57.1%; *n* = 36/63) than in benign diseases (13.2%; *n* = 5/38).

Research findings indicate that the expression of PSMA varies across different types of thyroid cancer. According to the study by Bychkov et al., the expression of PSMA was reported to be 19% in follicular adenomas, 46% (*n* = 24/52) in follicular thyroid carcinoma (FTC), and 51% in papillary thyroid carcinoma (PTC) (*n* = 61/120) [4]. These percentages were consistent with another study, revealing 40% (*n* = 4/10) in FTC and 58% (*n* = 18/31) in PTC. Santhanam et al. investigated the expression of PSMA in local metastatic thyroid cancer in lymph nodes. All three lymph node samples investigated stained positive for PSMA expression in the neovasculature [5]. 

In their exploration of PSMA expression as a biomarker for DTC aggressiveness and outcome prediction, Sollini et al. conducted a study involving 59 cases [6]. The study found that PSMA expression varied among the cases analyzed. The percentage of PSMA expression was ≤10% in 17 cases, 11 to 79% in 31 cases, and ≥80% in 11 cases. The multivariate analysis identified several significant predictors of distant metastases, including age, sex, histotype, vascular invasion, T and N parameters, and PSMA positivity. The final multivariate model that predicted RAI-R included the stage, high PSMA expression (≥80%), and the interaction between moderate PSMA expression (11 to 79%) and the stage. Ciappucino’s findings complemented this research by indicating that patients aged ≥ 55 years, and those with large primary tumors (>40 mm) or aggressive subtypes, had a higher mean immunoreactive score (IRS), which correlated with structural disease at the last follow-up [7]. Strong PSMA expression (IRS ≥ 9) was associated with a shorter progression-free survival (PFS). Bychkov et al. reported PSMA expression’s correlation with tumor size (*p* = 0.02) and vascular invasion in FTC (*p* = 0.03), while no significant associations were observed with other baseline histological and clinical parameters [4].

Feng et al. conducted a comprehensive investigation employing Western blot, PCR, and [68Ga]Ga-PSMA uptake experiments on cell lines, complemented by in vivo small animal imaging [8]. Their study revealed a significant correlation between ^68^Ga-PSMA imaging and tumor burden, as evidenced by correlations with ^18^F-FDG PET/CT (8.08 ± 7.74 and 5.67 ± 4.23, *p*  =  0.01) and thyroglobulin (Tg) levels (307.1  ±  183.4 vs. 118.0  ±  116.1, *p*  =  0.002). Notably, the research indicated an association between increasing PSMA expression and decreasing thyroid cancer differentiation. However, the ^68^Ga-PSMA uptake in thyroid cancer patients was not significantly linked solely to the degree of thyroid cancer differentiation but also demonstrated associations with the metabolic activity and tumor burden, as reflected by the ^18^F-FDG PET/CT and Tg levels. In a parallel study, Ciappuccini et al. investigated PSMA expression in neck persistent/recurrent disease (PRD) utilizing immunohistochemistry and its correlation with RAI or ^18^F-FDG uptake, along with patient outcomes in a cohort of 44 individuals [7]. The quantification of PSMA expression using the immunoreactive score (IRS) revealed that 68% of patients exhibited at least one PSMA-positive lesion (IRS ≥ 2), with comparable proportions in RAI-positive and RAI-negative patients (75% vs. 66%). In RAI-negative patients, however, the prevalence of PSMA-positive disease (79% vs. 25%, *p* < 0.01) and the mean IRS (4.0 vs. 1.0, *p* = 0.01) were notably higher in ^18^F-FDG-positive compared to ^18^F-FDG-negative patients.

Several imaging studies have been conducted using PSMA-based radiotracers such as ^68^Ga-PSMA-11 (68Ga-HBED-CC-PSMA) and ^18^F-DCFPyL (as shown in Table 1 and Table 2). The efficacy of ^68^Ga-PSMA-11 has been studied for various purposes such as detecting DTC metastases, identifying patients suitable for PSMA-targeted radionuclide therapy, diagnosing patients with RAI-R DTC, for patients with TENIS, and detecting patients with elevated Tg or anti-Tg antibodies.

In a case report by Verburg et al., strong uptake was seen in the cervical lymph nodes and pulmonary metastases in a patient with RAI-R DTC [9]. In a study by Verma et al. (2018) focusing on patients with iodine-avid metastatic disease, substantial PSMA uptake was observed in all the cases [10]. ^68^Ga-PSMA PET/CT detected 93.75% (*n* = 30/32) of the total lesions, with a median SUVmax of 31.35, outperforming ^18^F-FDG PET/CT, which was positive in 81.85% (*n* = 23/32) of the lesions. Notably, 70% of the lesions with PSMA expression were localized to the bones (*n* = 21/30).

A prospective study was conducted on patients with iodine-negative and FDG-positive metastasized DTC [11]. The study found that ^68^Ga-PSMA-11 PET/CT was able to successfully identify metastatic disease in five out of six patients, with all 42 lesions confirmed by ^18^F-FDG PET/CT or conventional CT imaging. Using the ^68^Ga-PSMA-11 PET/CT, all the tumor lesions identified with the ^18^F-FDG PET/CT imaging could be visualized in three of the five patients. In two patients, only the most prominent lesions detected with the ^18^F-FDG PET/CT imaging were visualized by the ^68^Ga-PSMA-11 PET/CT.

In a prospective pilot study by Verma et al., PSMA expression was detected in two TENIS patients using ^68^Ga-PSMA-11 PET/CT [10]. The lesions were found in the bones and lungs, and the ^68^Ga-PSMA PET/CT localized a lesion in each patient similar to the ^18^F-FDG PET/CT results. In a later study, PSMA expression was found in five out of nine patients with TENIS [12]. A total of fourteen lesions were detected on the CT scan, and the ^68^Ga-PSMA PET/CT detected nine of them (64.28%), with SUVmax ranging from 10.1 to 45.67 and median SUVmax of 16.31. On the other hand, ^18^F-FDG PET/CT was positive in 11 out of 14 lesions (78.57%). The lesions that showed PSMA uptake were localized in bones (five out of nine) and lungs (four out of nine). Two lesions localized in the iliac crest and acetabulum were missed on the ^18^F-FDG PET/CT but were seen on CT and the ^68^Ga-PSMA PET/CT. In a case study with one patient, ^68^Ga-PSMA-11 PET/CT was used to detect metastases in TENIS [13]. Intense radiotracer uptake was found in mediastinal and left supraclavicular lymph nodes, brain metastases, bilateral lung nodules, and skeletal sites. The patient also underwent ^18^F-FDG PET/CT, which demonstrated similar findings; however, the number of lesions detected in the brain was fewer compared to ^68^Ga-PSMA PET/CT. In the investigation of ^68^Ga-PSMA-11 for detecting DTC metastases, Lawhn-Heath et al. (*n* = 7) found a detection rate of 93.8% for 18F-FDG PET/CT and 53.1% for ^68^Ga-PSMA PET/CT [14]. The median lesion SUVmax was 9.0 for the ^18^F-FDG PET/CT (range 3.5–28.4) and 9.2 for the ^68^Ga-PSMA PET/CT (range 2.0–27.8). The ^68^Ga-PSMA PET/CT had the highest detection rate for FTC, reaching 80.0%. In another study (*n* = 10), ^68^Ga-PSMA-11 PET/CT successfully identified all 64 lesions, each of which was also confirmed by CT [15]. In contrast, ^131^I SPECT/CT detected 55 out of the 64 lesions, with discrepant lesions localized in the lung (44.4%), brain (22.2%), postoperative thyroid bed (11.1%), lymph nodes (11.1%), and bone (11.1%). The degree of agreement among the observers was notably high for both radiotracers. For the ^68^Ga-PSMA-11, the agreement had a κ of 0.98 (95% CI, 0.80–0.91), whereas for the ^131^I, it was slightly lower (κ = 0.86; 95% CI, 0.71–0.76).

In their study on RAI-R DTC, de Vries et al. (*n* = 5) observed tracer uptake suggestive of distant metastases in all ^68^Ga-PSMA-11 PET/CT scans [16]. Three patients underwent comparative imaging with ^18^F-FDG PET/CT, revealing additional lesions on ^68^Ga-PSMA PET/CT in two patients. Shi et al. prospectively enrolled 23 DTC and 17 RAI-R DTC patients [17]. The study found that the detection rates of DTC and RAI-R DTC were lower with ^68^Ga-PSMA-11 PET/CT than with 18F-FDG PET/CT (60.00% vs. 90.00% for DTC and 59.38% vs. 96.88% for RAI-R DTC). Immunohistochemistry also revealed that RAI-R DTC had significantly higher PSMA expression levels than DTC, but this difference did not correlate significantly with the SUVmax on ^68^Ga-PSMA-11 PET/CT.

In a separate study, Santhanam et al. (2020) investigated the localization of metastases with ^18^F-DCFPyL PET/CT in three patients with metastatic RAI-R DTC [5]. The following are the results of two patient scans using 18F-DCFPyl: In the first patient, the scan picked up activity in the retropharyngeal CT contrast-enhancing lymph node. The ^18^F-DCFPyl scan showed a greater SUV (3.1) compared to the ^18^F-FDG PET/CT (1.8). In the second patient, the ^18^F-DCFPyl scan showed intense uptake in the L3 vertebra, which was not visible on the post-treatment ^131^I whole-body scan or the ^18^F-FDG PET/CT. An MRI scan of the lumbar spine confirmed the presence of a sclerotic–lytic lesion in that location, indicating metastatic disease. Singh et al. also presented a case study, in which ^18^F-DCFPyl PET/CT was able to detect PTC [18]. 

## 3. Fibroblast Activation Protein Inhibitor-Based Radiotracers for Differentiated Thyroid Cancer

Fibroblast activation protein inhibitor (FAPI) is a relatively new tracer. In the pioneering study of Kratochwil (2019), ^68^Ga-FAPI-04 showed tracer uptake in 28 different kinds of cancer [19]. In recent years, ^68^Ga-FAPI imaging has been explored for various purposes in different clinical settings with promising results [20]. 

Fibroblast activation protein (FAP) is a type II transmembrane serine protease and is highly expressed in more than 90% of epithelial tumors; it is closely associated with tumor invasion and metastasis [21]. Using FAP as a target, different FAP inhibitors (FAPIs) have been developed. FAP is highly expressed on the cancer-associated fibroblasts’ membrane but not in normal tissue. It is thus associated with tumor invasion, metastasis, angiogenesis, and prognosis [22]. Its use has been investigated in the setting of detecting metastases for patients with RAI-R and patients with TENIS syndrome.

In the preclinical study conducted by Sun et al., it was observed that the presence of the BRAFV600E mutation correlated with heightened expression of cancer-associated fibroblast membrane proteins, including FAP [23]. Consequently, the use of FAPI may offer enhanced accuracy compared to ^18^F-FDG PET/CT, especially in patients with more aggressive disease manifestations.

Clinical studies have assessed the diagnostic accuracy of different FAPi-based radiotracers for different thyroid cancer indications. These tracers include ^68^-FAPI (not further specified), ^68^Ga-DOTA.SA.FAPi, ^18^F-FAPI-42, ^18^F-FAPI-46, and ^68^Ga-DOTA-FAPI-04 for indication of RAI-R metastatic lesions, TENIS syndrome, and DTC with elevated Tg or anti-Tg antibodies (Table 1 and Table 2).

Two case studies report on ^68^Ga-FAPI PET/CT for diagnosing metastases in patients with TENIS syndrome. One case study presents the first case of TENIS syndrome with FAPI-avid metastatic lesions. The ^68^Ga-FAPI PET/CT localized abnormal foci at the laryngeal mass and bilateral pulmonary nodules [24]. The second case study compares ^18^F-FDG PET/CT with FAPi: Compared with ^18^F-FDG PET/CT, the ^68^Ga-FAPI PET/CT showed more and higher metabolic lesions, including lesions in the liver, bones, and abdominal lymph nodes [25]. 

The study with the most patients included is a retrospective study (*n* = 117) focused on comparing ^68^Ga-DOTA.SA.FAPi and ^18^F-FDG PET/CT in patients’ RAI-R DTC. The ^68^Ga-DOTA.SA.FAPi exhibited a superior detection rate for lymph nodes, liver, bowel, and brain metastases compared to the ^18^F-FDG PET/CT [26].

Mu et al. investigated ^18^F-FAPI-42 in patients with biochemical elevations in Tg or anti-Tg antibodies (*n* = 42) [27]. FAPI-positive local recurrence showed the highest uptake intensity, followed by lymphatic, other site-associated (bone and pleura), and pulmonary lesions. All the positive lesions showed statistically higher uptake of ^18^F-FDG PET/CT than that of ^18^F-FAPI-42 (SUVmax, 2.6 versus 2.1; *p* = 0.026). However, the SUVmax of the ^18^F-FAPI-42 was higher than that of the ^18^F-FDG PET/CT in local recurrences and lymphatic lesions (SUVmax, 4.2 versus 2.9 and 3.9 versus 3.4, respectively; *p* > 0.05).

In a study by Nourbakhsh et al. (2024) involving 14 patients with TENIS syndrome, the performance of ^18^F-FAPI-46 was compared to ^18^F-FDG PET/CT imaging [28]. All the patients exhibited active lesions in both scans, with significantly lower SUVmax values in the liver and blood pool for the FAPI compared to the ^18^F-FDG PET/CT (*p* = 0.001). Notably, the standard SUV of the hottest lesion to the liver exceeded three in all FAPI scans but only in half of the ^18^F-FDG PET/CT scans, highlighting FAPI’s consistent identification of high metabolic activity lesions. In one case, the FAPI detected pulmonary nodules (SUVmax = 3.8) missed by the ^18^F-FDG PET/CT (SUVmax = 0.9), resulting in a 20% upstaging of patients.

In a study (*n* = 24) investigating ^68^Ga-DOTA-FAPI-04 PET/CT, positive lesions were identified in 87.5% of patients, particularly in lung and lymph node metastases [29]. In a case report, ^68^Ga-DOTA-2P(FAPI)2 was also able to detect PTC in a patient with diffuse lymph node metastases after total thyroidectomy and multiple cycles of RIT [30]. The tumor lesion became clearly visible with a good tumor-to-background ratio.

## 4. Arg-Gly-Asp-Based Radiotracers for Differentiated Thyroid Cancer

The tripeptide Arg-Gly-Asp (RGD) acid sequence has a high affinity and specificity for integrin αvβ3, a receptor found in various malignant tumors. RGD-based tracers have been developed for diverse imaging modalities, such as positron emission tomography, single photon emission computed tomography (SPECT), molecular magnetic resonance imaging, optical fluorescence, optical bioluminescence, photoacoustic imaging, and targeted contrast-enhanced ultrasound. RGDs have undergone testing in multiple cancer types, including melanoma, brain tumors, sarcoma, squamous cell carcinoma of the head and neck, glioblastoma multiforme, breast cancer, and rectal cancer. The versatility of RGD-based tracers reflects their potential utility across a spectrum of imaging strategies and cancer types.

The Arg-Gly-Asp (RGD) sequence has been identified for its ability to attach to the αvβ3 integrin found on the surface of angiogenic blood vessels or tumor cells. Integrin ανβ3 is essential for cell migration and invasion and plays an important role in tumor angiogenesis. Integrin ανβ3 expression is high on the surface of activated endothelial cells in newly formed blood vessels but is low in both resting endothelial cells and most normal organ systems, thus representing an interesting molecular marker for angiogenesis imaging. The RGD motif can be used to bind radiolabeled analogs. Thus, various radiolabeled derivatives of RGD peptides have been developed for angiogenesis imaging. Based on RNA-seq data from The Cancer Genome Atlas, the expression of αv and β3 integrin subunits is particularly high in thyroid cancer. Integrin αvβ3 overexpression in thyroid cancer is likely due to its presence on both the activated endothelial cells of the cancer neovasculature and the cancer cell surface, as shown by several reports documenting αvβ3 integrin expression in thyroid cancer cell lines [31]. For DTC, RGD-based radiotracers have been investigated for the detection of DTC lymph node metastases, RAI-R patients, and patients with a negative post-therapy ^131^I-scan.

The preclinical study of Cheng et al. (2016) investigated the antagonism of Arg-Gly-Asp (RGD)-binding integrin activity in three PTC cell lines (BCPAP, K1, and TPC1) [32]. The research revealed that all three cell lines exhibited moderate-to-high expression of RGD-binding integrin heterodimers αvβ3 and αvβ5. Antagonizing these integrins resulted in a significant inhibitory effect on cell viability, stronger apoptotic responses in TPC1 cells, and substantial inhibition of migration and invasion across all three PTC cell lines.

There are two clinical studies investigating the diagnostic accuracy of RGD-based radiotracers for the detection of DTC. These are ^18^F-AIF-NOTA-PRGD2 and ^68^Ga-DOTA-RGD2.

Cheng et al. (2014) (*n* = 20) compared ^18^F-AIF-NOTA-PRGD2 with ^18^F-FDG PET/CT for the detection of lymph node metastases of DTC, revealing that although ^18^F-AIF-NOTA-PRGD2 exhibited abnormal uptake in most DTC lymph node metastases, its diagnostic value was inferior to ^18^F-FDG PET/CT [33]. For all malignant lesions, the mean SUV for the ^18^F-FDG PET/CT was significantly higher than that for the ^18^F-AIF-NOTA-PRGD2 (*p* < 0.05). Another study aimed to compare the diagnostic accuracy of ^68^Ga-DOTA-RGD2 PET/CT with ^18^F-FDG PET/CT in RAI-R patients [34]. In a prospective enrollment of 44 RAI-R DTC patients, the ^68^Ga-DOTA-RGD2 PET/CT demonstrated an overall sensitivity of 82.3%, specificity of 100%, and accuracy of 86.4%, while the ^18^F-FDG PET/CT showed a sensitivity of 82.3%, specificity of 50%, and accuracy of 75%.

## 5. [^18^F]Tetrafluoroborate

In 2017, a new PET tracer was tested in healthy human subjects for the imaging of DTC: ^18^F-tetrafluoroborate (^18^F-TFB). This study showed promising results of using ^18^F-TFB PET/CT for noninvasive NIS imaging, as ^18^F-TFB can mimic iodide transport. ^18^F-TFB can go through the NIS receptor and gets trapped in thyroid cells.

^18^F-TFB has diagnostic potential in detecting (recurrent) differentiated thyroid cancer and cervical lymph node metastases. One indication for an ^18^F-TFB PET/CT scan is the detection of pre-operative cervical lymph node metastases to avoid a second surgery. Another indication is to predict whether radioiodine therapy will be successful in a patient. ^18^F-TFB PET/CT has three advantages over ^123^I- or ^131^I-scintigraphy. First of all, ^18^F-TFB is a positron emitter detected by PET, which is more sensitive than a gamma camera. Second, the scan takes place on the same day as the administration of the tracer, so patients do not have to attend an extra day (as is the case in iodine-123(131) scintigraphy). Third, it has a lower radiation dose.

Pre-clinical studies have been performed in xenograft mice and cell lines. These studies concluded that ^18^F-TFB has specific uptake by the sodium/iodide symporter. Jarugeui-Osoro et al. showed a rapid accumulation of ^18^F-TFB in FRTL-5 rat thyroid cells and in vivo in the thyroid and stomach [35]. This was confirmed by Weeks et al. and Khoshnevisan et al. in NIS-expressing human colon cell [36,37]. Diocou et al. found that ^18^F-TFB was superior to I-123 due to better pharmacokinetics (faster tumor uptake and faster and more complete clearance from circulation) [38]. Niu et al. illustrated the possibilities of ^18^F-TFB in NIS-expressing cell lines for the diagnosis of stomach cancer [39]. Goetz et al. provided further information on the time–activity curves and fitted kinetics for ^18^F-TFB in mice [40]. Jiang et al. assessed the influence of specific activity on tumor uptake in NIS-expressing C6-glioma xenograft tumors [41]. Marti-Clement et al. conducted whole-body PET imaging in two non-human primates. It concluded that it is a very useful radiotracer in primates, with a characteristic biodistribution in organs expressing NIS [42].

In four clinical phase I studies with a total of 47 patients, ^18^F-TFB PET/CT proved to be safe and effective in humans [43,44]. In these clinical trials, the safety, pharmacokinetics, metabolism, biodistribution, and dosimetry were assessed [44]. ^18^F-TFB has a short half-life of 110 min, low-energy (Emax = 0.634 MeV), and high-yield (97%) positron emission [43]. In human subjects, it showed rapid renal clearance and no uptake in bone and joints, meaning that ^18^F-fluoride was not released in vivo after IV administration [43]. Next to this, ^18^F-TFB imaging was compared to ^124^I PET/CT, ^18^F-FDG PET/CT, and ^131^I scintigraphy [43,45]. The ^18^F-TFB PET/CT showed higher sensitivity and accuracy than ^131^I in whole-body scintigraphy and SPECT-CT. The ^18^F-TFB PET/CT was not inferior to the ^124^I PET/CT in detecting thyroid cancer and was able to detect metastases that were not found on the ^124^I PET/CT [43].

## 6. Discussion

Studies have suggested that PSMA PET/CT is better than ^18^F-FDG PET/CT in detecting lesions in bones and lungs. These radiotracers have also been linked to radioiodine-refractory thyroid cancer. Additionally, PSMA expression levels have been explored as potential biomarkers for predicting distant metastases and radioiodine therapy (RIT) outcomes, and their inclusion in multivariate models for predicting RAI-R highlights their significance in risk stratification. FAPI-based radiotracers have also shown potential in predicting and assessing RAI-R, with higher detection rates in specific metastatic sites like lymph nodes and the liver compared to ^18^F-FDG PET/CT. This demonstrates their value as a complementary tool in evaluating refractory disease. Furthermore, RGD-based radiotracers have shown diagnostic accuracy in patients with RAI-R-differentiated thyroid cancer (DTC), with their sensitivity and specificity indicating their potential role in predicting and confirming RAI-R, particularly when compared to ^18^F-FDG PET/CT. ^18^F-TFB is proving to be a promising PET tracer for noninvasive NIS imaging in DTC. Its ability to mimic iodide transport and provide diagnostic potential for detecting cervical lymph node metastases offers advantages over traditional scintigraphy methods. The development of targeted radionuclide therapies based on these tracers shows promise in advancing the management of refractory disease. Future research should focus on larger clinical trials, the standardization of imaging protocols, and head-to-head comparisons to establish the most effective and reliable tracers for specific clinical scenarios.

Several limitations should be acknowledged in this study. Firstly, the scope of the available literature concerning the tracers and indications under review is constrained by a limited number of studies and the limited number of inclusion of patients within these studies. Secondly, despite efforts to comprehensively review the English literature, it is possible that some relevant studies may have been missed. Moreover, the heterogeneity observed within the included studies poses a significant challenge. The variability in patient populations, study designs, imaging protocols, and outcome measures across different studies hinders the possibility of performing a meta-analysis and thus limits the establishment of definitive conclusions.

In conclusion, the use of various radiotracers in differentiated thyroid cancer is a rapidly evolving field with promising potential for improving diagnostic accuracy, patient stratification, and treatment selection. The current evidence underscores the preferential utility of select tracers. Notably, ^18^F-TFB demonstrates promise in enhancing imaging resolution for DTC staging. Conversely, in the context of advanced potentially dedifferentiated DTC, tracers targeting FAP display encouraging preliminary efficacy. PSMA and RGD, in contrast, show a comparatively disappointing outcome. Hence, for future research, the authors recommend investigating FAP-targeting tracers in the advanced setting and ^18^F-TFB for staging DTC, rather than further investigating PSMA and RGD.

**Table 1 cancers-16-01401-t001:** Imaging Studies.

Tracer Group	Tracer	Study	Indication	Number of Patients
PSMA	^68^Ga-PSMA-11	Verburg et al. (2015) [9]	RAI-R	1
		Taywade et al. (2016) [13]	TENIS	1
		Lutje et al. (2017) [11]	Iodine-negative and FDG-positive metastasized DTC	6
		Verma et al. (2018) [12]	DTC metastases + find patients suitable for PSMA-targeted radionuclide therapy	10
		Lawhn-Heath et al. (2020) [46]	Metastatic DTC	7
		De Vries et al. (2020) [16]	RAI-R	5
		Verma et al. (2021) [12]	TENIS, find metastatic lesions	9
		Pitalua-Cortes et al. (2021) [15]	Metastatic DTC	10
		Shi et al. (2023) [17]	RAI-R	40
	^18^F-DCFPyL	Singh et al. (2018) [18]	NS	1
		Santhanam et al. (2020) [5]	Elevated Tg	2
FAPI	^68^Ga-FAPI	Fu et al. (2021) [24]	RAI-R metastatic lesions	1
		Fu et al. (2021) [24]	TENIS syndrome	1
		Wu et al. (2021) [25]	TENIS syndrome	1
	^68^Ga-DOTA.SA.FAPi	Ballal et al. (2022) [26]	RAI-R metastatic lesions	117
	^18^F-FAPI-42,	Mu et al. (2023) [27]	DTC with elevated Tg or anti-Tg antibodies	27
	^18^F-FAPI-46	Nourbakhsh et al. (2024) [28]	TENIS syndrome	14
	^68^Ga-DOTA-FAPI-04	Chen et al. (2022) [29]	RAI-R metastatic lesions	24
		Tatar et al. (2023) [30]	DTC metastases	1
	^68^Ga-DOTA-2P(FAPI)2	Zhao et al. (2022) [47]	PTC with LNM	1
RGD	^18^F-AIF-NOTA-PRGD2	Cheng et al. (2014) [33]	Lymph node metastases of DTC	20
	^68^Ga-DOTA-RGD2	Parihar et al. (2020) [34]	RAI-R, patients with negative post-therapy 131I-scan.	44
TFB	^18^F-TFB	O’Doherty et al. (2017) [48]	DTC	5
		Samnick et al. (2018) [43]	DTC + LNM	9
		Dittmann et al. (2020) [45]	Local recurrence + metastases	25

**Table 2 cancers-16-01401-t002:** Detection Rate of Different Tracers for Thyroid Cancer Indications.

Indication Thyroid Cancer	Tracer	Results	Conclusions
Recurrent DTC	^18^F-TFB	52% detection rate (^131^I WBS detection rate = 12%) [45]	^18^F-TFB PET detected local recurrence or metastases of DTC in significantly more patients than conventional 131I-dxWBS and SPECT-CT.
Detection of DTC metastases	^68^Ga-PSMA-11	53.1% detection rate (FDG = 93.8% detection rate) [46]; 100% detection rate (Pitalua-Cortes, Garcia-Perez et al., 2021); 100% detection rate [10]; 60% detection rate (FDG = 90% detection rate) [17]	^68^Ga-PSMA-11 PET/CT can detect thyroid cancer metastases, but its detection rate is lower than that of 18F-FDG PET/CT.
	^68^Ga-DOTA.SA.FAPi	95.4% detection rate lymph nodes (FDG = 86.6%) [49]	^68^Ga-DOTA.SA.FAPi can detect lymph nodal, liver, bowel, and brain metastases better than ^18^F-FDG in patients with RAI-R DTC.
	^68^Ga-DOTA-FAPI-04	87.5% detection rate [29]	^68^Ga-DOTA-FAPI-04 PET/CT has a promising detection rate for RAI-R DTC metastasis.
	^68^Ga-DOTA-RGD2	86.4% diagnostic accuracy (FDG = 75% diagnostic accuracy) [34]	^68^Ga-DOTA-RGD2 PET/CT showed similar sensitivity to, but higher specificity and overall accuracy than ^18^F-FDG PET/CT in detection of lesions in RAI-R DTC.
TENIS	^68^Ga-FAPI	100% detection rate [50]	First case of TENIS with FAPI-avid metastatic lesions.
	^68^Ga-PSMA-11	100% detection rate [13]; 64.28% detection rate (FDG = 78.57% detection rate) [12]	^68^Ga-PSMA-11 PET/CT demonstrates PSMA expression in TENIS patients with lesions being localized to the bones, lungs, mediastinal, and left supraclavicular lymph nodes, brain, and bilateral lung nodules.
Lymph node metastases of DTC	^18^F-AIF-NOTA-PRGD2	FDG > RGD [33]	No correlation was found between the uptake of ^18^F-AIF-NOTA-PRGD2 and 18F-FDG, which may suggest the two tracers provide complementary information in DTC lesions.
	^18^F-DCFPyL	100% detection rate [5]	^18^F-DCFPyl may prove useful for the localization of metastases in patients with metastatic RAI-refractory DTC.
	^18^F-TFB	100% detection rate [43]	^18^F-TFB PET was not inferior to ^124^I-PET in detecting thyroid cancer and its metastases and was able to detect ^124^I-PET-negative viable differentiated thyroid cancer metastases.

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
