# Peer review of "Advances in the Development of Positron Emission Tomography Tracers for Improved Detection of Differentiated Thyroid Cancer"

_cancers, 2024, doi:10.3390/cancers16071401_

Round 1
Reviewer 1 Report
Comments and Suggestions for Authors
There are major issues that need to be amended, revised, and fixed. The comments and suggestions are as stated below.
Introduction:
- The author should critically discuss the main focus of the study.
- Please elaborate more on the background of the study and the improvement of current research compared to previous research.
Results and Discussion:
- The discussion parts in this section are very shallow. The authors should include their critical opinion and related previous studies/references on state of the art in the manuscript.
- The authors should cite the references in Tables 1 and 2. Please revise to include a suitable standard for citing this reference.
Conclusion:
i. The conclusion part is missing. This is the important section in which the authors should conclude their findings, especially for the review paper.
ii. The authors should conclude the main finding in this section.
iii. Describe the study's limitations and future improvement by including the data provided by the authors.
Comments on the Quality of English Language
Minor editing of English language required.
Author Response
Thank you for taking the time to review or article on novel PET tracers for differentiated thyroid cancer. We have adapted the manuscript accordingly.
- The author should critically discuss the main focus of the study. – We have expanded the aim of the study. “In recent times, much effort has been spent on trying out numerous new tracers for imaging in various clinical stages of DTC, in order to advance imaging sciences especially there where ultrasound and 18F-FDG PET/CT are not sufficient. This has resulted in numerous publications on various situations, without a good overview of the performance of different tracers in similar situations currently available in literature. Hence, this review aims to comprehensively evaluate the literature on new tracers for different clinical indications in thyroid cancer, such detecting lymph node metastases, TENIS (thyroglobulin elevation/negative iodine scintigraphy) syndrome, recurrent or persistent disease, and distant metastases in DTC.”
- Please elaborate more on the background of the study and the improvement of current research compared to previous research. – As this is an invited review on the topic of novel tracers for DTC, we are unsure on how to further expand on this.
- The discussion parts in this section are very shallow. The authors should include their critical opinion and related previous studies/references on state of the art in the manuscript/ The conclusion part is missing. This is the important section in which the authors should conclude their findings, especially for the review paper./ The authors should conclude the main finding in this section. – We have now added clear statements on which tracers are preferable for detection of DTC, to the discussion and to the conclusion. In short: given the promising staging capabilities of 18F-TFB and the efficacy of FAP-targeting tracers in advanced, potentially de-differentiated cases, continued investigation in these domains is justified.
- The authors should cite the references in Tables 1 and 2. Please revise to include a suitable standard for citing this reference. – We have changed the references for the tables accordingly.
- Describe the study's limitations and future improvement by including the data provided by the authors. – We have added the limitations and future recommendations to the discussion. “Several limitations should be acknowledged in this study. Firstly, the scope of available literature concerning the tracers and indications under review is constrained by a limited number of studies and the limited number of inclusion of patients within these studies. Secondly, despite efforts to comprehensively review the English literature, it is possible that some relevant studies may have been missed. Moreover, the heterogeneity observed within the included studies poses a significant challenge. The variability in patient populations, study designs, imaging protocols, and outcome measures across different studies hinders the possibility of performing a meta-analysis, and thus limits the establishment of definitive conclusions.”
Reviewer 2 Report
Comments and Suggestions for Authors
I have reviewed the article titled "Advances in the Development of Positron Emission Tomography Tracers for Improved Detection of Differentiated Thyroid Cancer." The article aims to explore the use of radiotracers in the diagnosis and management of thyroid cancer, specifically prostate-specific membrane antigen (PSMA)-based radiotracers, fibroblast activation protein inhibitor (FAPI)-based radiotracers, Arg-Gly-Asp (RGD)-based radiotracers, and 18F-tetrafluoroborate (18F-TFB).
PSMA-based radiotracers, originally developed for prostate cancer imaging, have shown potential in detecting thyroid cancer lesions but have a lower detection rate than 18F-FDG PET/CT. FAPI-based radiotracers, targeting fibroblast activation protein highly expressed in tumors, offer potential in the detection of lymph nodes and radioiodine-resistant metastases. RGD-based radiotracers, binding to integrin αvβ3 found on tumor cells and angiogenic blood vessels, demonstrate diagnostic accuracy in detecting radioiodine-resistant thyroid cancer metastases.
18F-TFB emerges as a promising PET tracer for imaging lymph node metastases and recurrent DTC, offering advantages over traditional methods. The purpose of my review reports to provide constructive feedback on the manuscript and suggest revisions that the authors should address before finalizing the manuscript.
1. I understand that the authors failed to make their article appealing to the nuclear medicine audience. To do so, they need to present their data and figures in a more engaging and accessible way, beyond technical details. This will help attract readers who are not experts in the field.
2. One suggestion is to include graphic figures of imaging studies. Obtaining copyright permission for these figures should be easy and may help attract the interest of the general audience of the journal.
3. It is also worth noting that a similar review has already been published (DOI https://www.mdpi.com/2072-6694/13/19/4748; https://www.mdpi.com/1420-3049/27/15/4936).
4. Finally, I noticed that there is a significant amount of plagiarism/percent matches around 45%. It is recommended to reduce this to around 20%.
Comments on the Quality of English LanguageModerate editing of English language required
Author Response
Thank you for taking the time to review or article on novel PET tracers for differentiated thyroid cancer. We have adapted the manuscript accordingly.
- I understand that the authors failed to make their article appealing to the nuclear medicine audience. To do so, they need to present their data and figures in a more engaging and accessible way, beyond technical details. This will help attract readers who are not experts in the field. – We would be happy if the reviewer could give one or more examples how we can achieve this. We have tried to write about a very technical subject as accessible as possible. If there are any recommendations on how to make it more accessible we would be happy to change the manuscript accordingly.
- One suggestion is to include graphic figures of imaging studies. Obtaining copyright permission for these figures should be easy and may help attract the interest of the general audience of the journal. – Due to the deadline of the revisions it has not been possible to achieve this.
- It is also worth noting that a similar review has already been published (DOI https://www.mdpi.com/2072-6694/13/19/4748; https://www.mdpi.com/1420-3049/27/15/4936). – We have read this review to make sure we included all the relevant studies. However, as this was an invited review, the authors did not want to deviate too much from the proposed topic.
- Finally, I noticed that there is a significant amount of plagiarism/percent matches around 45%. It is recommended to reduce this to around 20%. – We have conducted a plagiarism check as well; however no plagiarism was detected (grammarly.com).
Reviewer 3 Report
Comments and Suggestions for Authors
The manuscript by H.I. Coerts et al. submitted to Cancers described the development of PET imaging agents for detecting differentiated thyroid cancer, including PSMAI-based, FAPI-based, RGD-based radiotracers and 18F-TFB. The manuscript is well written and interesting. The interpretations and conclusions are justified. The manuscript seems to be of broad interest for chemist and biologist working in the area. Therefore, it is recommended for publication with the following issues being addressed appropriately.
Some revisions are as follows:
1. Page 2, line 66, “68Ga-PSMA PET/CT is now being used in clinical practice and trials for prostate cancer treatment.” should be corrected as “68Ga-PSMA PET/CT is now being used in clinical practice and trials for prostate cancer diagnosis.”
2. Page 3, line 119, in vivo should be in vivo.
3. Page 3, line 120, 68Ga-PSMA should be 68Ga-PSMA.
4. Page 5, line 161, PET/CTresults should be PET/CT results.
5. Page 6, line 207, -Ga-FAPI should be 68Ga-FAPI.
6. Page 8, line 274, he RGD should be the RGD.
7. Page 8, line 276, b3 should be b3.
8. Page 8, line 277, avb3 should be avb3.
9. Page 8, line 282, 131I-scan should be 131I-scan.
10. In the conclusion section, if the authors could forecast future directions of the PET imaging agents for detecting differentiated thyroid cancer, it would be helpful to the readers.
11. In particular, there are several format issues in References section.
Comments on the Quality of English Language
None
Author Response
Thank you for taking the time to review or article on novel PET tracers for differentiated thyroid cancer. We have adapted the manuscript accordingly.
- tm 9 were formatting suggestions, we have changed the sentences.
- In the conclusion section, if the authors could forecast future directions of the PET imaging agents for detecting differentiated thyroid cancer, it would be helpful to the readers. - We have now added clear statements on which tracers are preferable for detection of DTC, to the discussion and to the conclusion. In short: given the promising staging capabilities of 18F-TFB and the efficacy of FAP-targeting tracers in advanced, potentially de-differentiated cases, continued investigation in these domains is justified.
- In particular, there are several format issues in References section. - We have changed the references accordingly
Round 2
Reviewer 2 Report
Comments and Suggestions for Authors
The authors performed significant revisions based on the reviewer's comments, the MS can be acceptable for publication.
Comments on the Quality of English LanguageMinor editing of English language required